# Effects of Visual Cue Deprivation Balance Training with Head Control on Balance and Gait Function in Stroke Patients

**DOI:** 10.3390/medicina58050629

**Published:** 2022-04-30

**Authors:** Seung-Min Nam, Do-Youn Lee

**Affiliations:** 1Department of Sports Rehabilitation & Exercise Management, Yeungnam University College, Daegu-si 42415, Korea; ngd1339@naver.com; 2Department of Physical Therapy, Daegu University, Gyeongsan-si 38541, Korea

**Keywords:** visual cue deprivation balance training, head control, balance, gait, stroke

## Abstract

*Background and Objectives:* Visual cue deprivation is the instability of head control is increased. The purpose of this study is to investigate the effects of visual cue deprivation balance training by applying head control feedback to the balance and gait ability of stroke patients. *Materials and Methods**:* The study was conducted on 41 patients diagnosed with hemiplegia due to stroke. Subjects were randomly assigned to any of the following groups: the experimental group I, the experimental group II or the control group. The randomization method used a simple randomization method. To evaluate changes in balance function, a LOS (Limit of Stability) and a BBS (Berg Balance Scale) were performed. In addition, to evaluate changes in ST (stride time), SL (stride length), and cadence, a LEGSys were performed. *Results:* A two-way repeated ANOVA was conducted to analyze the differences between groups. There were significant differences between groups in all variables for the balance function. There were significant differences between groups in all variables for the balance function. There were significant differences between groups in SL and cadence for the gait function. *Conclusions:* Visual cue deprivation balance training applying head control feedback is effective in improving dynamic balance ability and cadence. It is necessary to constantly maintain the head orientation by feedback and to properly control the head movement.

## 1. Introduction

A stroke is defined as a neurological defect caused by damage to blood vessels in the brain [1]. A stroke generally causes various impairments in motor function [2]. Hemiplegia due to stroke has problems such as asymmetric posture, unstable body balance, decreased weight transfer ability, and decreased walking ability, which increases the risk of falls and delays return to daily life [3].

Balance is achieved by integrating afferent stimuli input from vestibular, visual, auditory, proprioceptors, and sensory receptors in the central nervous system [4]. The problem of balance ability decreases movement, delays recovery of activities in daily life, increases the risk of falls, and acts as a factor that decreases standing ability and gait ability [5]. Balance in a standing position is closely related to the ability to transfer weight to either lower extremity [6]. This is a prerequisite for performing functional movements or activities in daily life and is important for various activities such as sitting to stand, gait, changing direction, and moving up and down stairs [7].

Gait is a complex process involving the body, and it is related to balance and postural control ability, and the integration of sensory input is very important to maintaining balance during walking [8,9]. However, in stroke patients, it is difficult to integrate sensory input due to the deterioration of proprioceptive sense, and it is difficult to maintain balance due to partial weight support, and the gait ability and performance independent of daily activities deteriorate [10].

Neurodevelopmental approaches, task-oriented training, and visual cue deprivation training are known as training for restoring balance and gait disorders in recent stroke patients [11]. To maintain a normal gait and constant balance, it is necessary to control the following three different sensory information: visual, somatosensory, and vestibular sensory. However, in stroke patients, the role of visual information is dominant, and excessive visual information interferes with balance and gait ability [12]. Therefore, visual cue deprivation can reduce visual dependence by promoting somatosensory and vestibular stimulation. The visual cue deprivation training has the characteristic of using somatosensory and vestibular sensation input appropriately by suppressing excessive visual dependence to improve the balance ability of stroke patients [13]. In other words, visual cue deprivation training is a treatment that induces the stroke patient’s proprioceptive somatosensory and vestibular sensory input [14].

However, since visual cue deprivation can give a feeling of psychological anxiety to stroke patients when performing balance training, there is a limitation that there is a high risk of self-exercising [15]. In addition, with visual cue deprivation training, it is difficult to maintain head orientation in space due to the cue deprivation of visual information, and the instability of head control is increased. In other words, there is a problem that the head stabilization strategy, which is one of the balance strategies, cannot be used [16]. Proper head orientation can create conditions for the most functional activities of the vision [17]. Moreover, according to a previous study, it was reported that because of performing vestibular stimulation training for stroke patients, and it was effective in improving the balance function. That is, the afferent information from the vestibular system maintains balance and has the function of preventing an imbalance of posture [18].

Therefore, it is of clinical significance to improve the balance ability of stroke patients. However, although studies on training methods for improving balance and gait ability of stroke patients are needed from a multilateral level, studies that prove the effectiveness of the visual cue deprivation training for maintaining head orientation in stroke patients are insufficient [19,20]. Therefore, the hypothesis of this study is that visual cue deprivation balance training applied to head control feedback will influence the balance and gait function of stroke patients. The purpose of this study is to investigate the effect of the visual cue deprivation balance training by applying head control feedback to the balance and gait ability of stroke patients.

## 2. Materials and Methods

### 2.1. Participants

The study was conducted on 45 patients diagnosed with hemiplegia due to stroke by a rehabilitation medicine specialist at Gyeongsan S Hospital. The purpose of the study and the contents of the experiment were explained to the subjects and voluntary consent was obtained. Based on the previously reported large effect sizes of visual cue deprivation balance training [21], we calculated a priori power in G Power 3.1.9.6 program by considering the use of the F test with 0.4 as estimated effect size, an alpha of 0.05, and 0.8 as statistical power. Total sample size was estimated at 39 subjects in this study. We selected a total of 45 subjects in consideration of the dropout rate. Finally, 4 dropped out due to discharge or treatment refusal, and a total of 41 completed the final experiment. The detailed selection criteria for subjects are as follows: Those who were diagnosed with hemiplegia by a rehabilitation medicine specialist, those who had been diagnosed with a stroke for 6 months, those who had an MMSE (Mini-Mental State Examination) score of 24 or higher, who could understand the research procedure, and those who could stand or gait independently were selected. This research was approved by the Daegu University Bioethics Committee (1040621-202007-HR-013).

### 2.2. Study Procedure

Subjects were randomly assigned to either a group visual cue deprivation balance training with head control (Experimental group I; *n* = 14) or a group visual cue deprivation balance training (Experimental group II; *n* = 13) or a group without visual cue deprivation (Control group; *n* = 14). The randomization method used a simple randomization method using a random number table. In addition, all evaluations were conducted using a single-blind, randomized, controlled design that anonymized the patients randomly assigned to the three groups. The training was conducted for 30 min three times a week for four weeks, depending on the training methods of each group. Experimental group I (EGI) and experimental group II (EGII) groups visual cue deprivation using eye bands. When using the eye bands from the beginning, the patient may feel dizziness, so balance training was conducted without the use of eye patch for 10 min of training. Moreover, balance training was performed in a state in which visual information was completely blocked by wearing an eye band for 20 min. Additionally, EGI provided passive feedback through verbal or body contact with the therapist to maintain head orientation. When head orientation in various directions was not maintained, the therapist instructed the patient to straighten the head verbally or straightened the patient’s head by hand. While control group performed balance training with their eyes open without visual cue deprivation. For balance training, training such as heel-toe standing, one-leg standing, trunk rotation, sit-to-stand, and stand-to-sit were conducted using a flat surface or a balance pad. All three groups were trained by the same therapist with more than 5 years of neurological treatment experience (Figure 1).

### 2.3. Static Balance Test

Static balance ability was measured using the Biorescue biofeedback analysis system (Marseille, France). It measures length (mm) and average speed (cm/s). To evaluate the static balance ability, the limit of stability (LOS) in the standing posture was measured. For the eight directions indicated on the monitor, the total distance and area of the center of mass were measured for the weight movements forward, back, left, and right. The monitor displayed an explanation of the measurement method and a demonstration of the test. The subject held one leg forward at 30° in a forward-looking standing position and then used an ankle strategy to achieve maximum range without losing balance. The limit of their ability to move from their center of mass was measured. However, if the foot falls off the platform or if the balance is disturbed and a therapist is supervising, the measurement will be repeated.

### 2.4. Dynamic Balance Test

Dynamic balance ability was measured using the Berg Balance Scale (BBS). This evaluation tool consists of a total of 14 activities related to balance, such as sitting, standing, and standing on one leg. For each task, the degree of performance was converted into a score on a 5-point scale from 0 to 4 points. If the task cannot be performed score is 0, and a maximum of 4 points is applied to each item if it is independently performed. The total score that can be obtained is 56 points, and a higher score means better balance. A score of 0–20 points indicates balance disorder, while a score of 41–56 points indicates good balance. Test-retest reliability for total scores was ICC = 0.92, interrater reliability was ICC = 0.97 [20].

### 2.5. Gait Function Test

LEGSys locomotion Evaluation and Gait System (Newton, MA, USA) is designed with lightweight and low-power embedded sensor technologies to be wearable comfortably during gait measurements. Five wearable sensors connected to the computer via Bluetooth, including a gyroscope, accelerometer, and magnetometer in three axes. Each sensor was attached with a strap to the anterior aspect of the tibia 3 cm above the ankle, the anterior aspect of the thigh 3 cm above the knee, and the center of the posterior superior iliac spine (PSIS). The sampling frequency used in this study was set at 100 Hz. Subjects were instructed to walk 7 m with at least 5 strides. Except for the first and last steps, the kinematic data, and spatiotemporal data of the middle three steps were recorded. Therapist supervised them from the side. In this study, stride time (s), stride length (m), and cadence of stroke patients were evaluated.

### 2.6. Statistics Analysis

The Shapiro–Wilk test was performed to check the normal distribution of each measured value, and all measured values satisfied normality. The data were presented as mean ± standard deviation (Mean ± SD), and the general characteristics of the subjects were presented as descriptive statistics. Two-way repeated ANOVA was conducted to analyze the differences between groups and measurement periods of the experimental group I, experimental group II, and control group. If the interaction was significant, a post hoc independent sample t-test was performed. A post hoc performed Bonferroni’s multiple comparison test. All results were considered statistically significant at *p* < 0.05. Differences among groups are indicated by letters such as a, b, and c. The data collected for this study were statistically processed using SPSS 22.0 for Windows (New York, NY, USA).

## 3. Results

There was no statistically significant difference in the general characteristics of the subjects between the three groups (Table 1). All three groups of subjects were chronic stroke patients with onset times of 25.37 in EG I, 23.28 in EG II, and 21.86 months in CG. For most patients, the treatment compliance rate increased as the treatment period elapsed. However, at the initial stage of treatment, there were adverse effects of dizziness and difficulty in maintaining balance as visual information was blocked.

There were significant differences between groups in all variables for the balance function (Table 2, Figure 2). The results reveal a significant interaction effect of group and time on LOSL (*F* = 11.103, *η*^2^ = 0.525), LOSR (*F* = 8.315, *η*^2^ = 0.455), LOSF (*F* = 9.057, *η*^2^ = 0.475), LOSB (*F* = 8.709, *η*^2^ = 0.467), and BBS (*F* = 20.433, *η*^2^ = 0.672).

There were significant differences between groups in SL and cadence for the gait function (Table 3, Figure 3). The results show that the interaction effects of group and time have a substantial interaction impact on SL (*F* = 17.608, *η*^2^ = 0.638) and cadence (*F* = 3.953, *η*^2^ = 0.286).

## 4. Discussion

Stroke patients do not use all their somatosensory, vestibular sensation, and vision to maintain balance, unlike normal adults, but mainly rely on vision to maintain balance [21]. This is because the factors of vestibular sensation and somatosensory sensation among afferent sensation inputs decrease, while the sensory stimulus dependent on vision increases, which has a dominant effect on maintaining balance [13]. However, the strategy of maintaining balance through excessive visual dependence may negatively affect balance and gait ability by acting as a hindrance to the use and integration of somatosensory and vestibular sensations [22]. Therefore, according to a recent study, it has been reported that the visual cue deprivation balance training is effective in improving the balance and gait ability of stroke patients [23]. However, since visual cue deprivation balance training does not properly maintain the head orientation of a stroke patient, there is a problem that it may interfere with vestibular sensation input [24]. Therefore, the purpose of this study was to investigate the effect of the visual cue deprivation balance training by applying head control feedback to the balance and gait ability of stroke patients.

To evaluate changes in static balance ability, a limit of stability was performed. The limit of stability increased significantly in the EGI and EGII after training, but no significant difference was observed in the CG. There was a significant difference between the three groups after training. To explain the difference between groups, the results of the post hoc analysis showed that EGI and EGII were significantly improved than CG. As a result of measuring the balance ability through 4 weeks of training, divided into the visual cue deprivation group and the visual tolerance group for stroke patients. These findings are consistent with the results of previous studies that showed visual cue deprivation training improves balance skills in stroke patients by allowing them to use somatosensory and vestibular sensation input correctly while suppressing excessive visual dependency [13]. In addition, it has been reported that visual block training is effective in improving the sense of joint position [25]. When visual information is blocked, the task must be performed by relying on the perceptual sense of and it can promote proprioceptive sensation activity [26]. In other words, balance is the ability to symmetrically distribute weight on both sides of the body and to maintain a posture or move without falling [27], and visual information plays a primary role in maintaining this balance. Therefore, stroke patients show excessive dependence on visual information, so it is considered that appropriate blocking is necessary [28].

To evaluate changes in dynamic balance ability, BBS was performed. Balance dysfunction is common in stroke patients. The Berg Balance Scale (BBS) is useful for evaluating the balance function of stroke patients, and it can estimate the minimal clinically important difference (MCID) in balance. According to previous studies, the MCID for improvement in balance as measured by the BBS was 13.5 points, indicating that the MCID does clinically detect changes in balance abilities in persons with stroke [29]. The BBS score increased significantly in the EGI after training, but no significant difference was observed in the EGII and CG. There was a significant difference between the three groups after training. To explain the difference between groups, the results of the post hoc analysis showed that EGI was significantly improved than EGII and CG. This result is thought to be due to additionally providing feedback on head control when performing visual cue deprivation training in EGI. According to previous studies, the head stabilization strategy is one of the strategies for maintaining balance as a method to continuously maintain the position of the head in space regardless of the movement of the trunk [30]. In general, vision recognizes objects in space and detects movement. Somatosensory makes it possible to recognize the positions of the head, trunk, and extremities in space based on sensory information received from muscle, joint, and skin receptors. The vestibular sense plays a role in transmitting information to the cerebrum about body movement and perception by encoding the head position, tilt, and angle [12]. In other words, the head stabilization strategy is one of the strategies for maintaining balance as a method of continuously maintaining the head orientation in space regardless of the movement of the trunk. Moreover, the function of the head and neck is to set the body’s standard for the surrounding environment and provide a stable basal surface for the head, visual system, and vestibular system during postural adjustment [31]. In addition, ankle strategy, hip strategy, and step strategy are used to maintain balance [32]. In particular, the hip strategy is mainly used when the center of gravity is large due to a response dependent on vestibular information, when moving quickly near the limit of stability, or when the mechanism is narrow or it is difficult to use the ankle strategy [33]. When hip joint movement occurs, the eyes move up and down to move away from the desired visual target to increase the input of the vestibular sensation [34]. That is, the dynamic balance ability maintains balance through the hip strategy. Therefore, when these previous studies and the results of this study are combined, the visual cue deprivation training that applies head control feedback is thought to be effective in enhancing the dynamic balance ability by promoting the input of the vestibular sensation of stroke patients.

To evaluate changes in gait ability, using LEGSys, stride time, stride length, and cadence were measured. The stride length, and cadence increased significantly in the EGI and EGII after training, but no significant difference was observed in the CG. There was a significant difference between the three groups after training. To explain the difference between groups, the results of post-hoc analysis showed that EGI and EGII were significantly improved than CG. According to previous studies, it was reported that balance training through visual cue deprivation was more effective in improving the gait ability of stroke patients compared to the visually dependent group [16]. In addition, it was reported that functional variables such as walking and climbing stairs were improved as a result of performing visual cue deprivation balance training on stroke patients, which was consistent with the results of this study [13]. Visual cue deprivation balance training improves motor learning and posture control on the affected side by activating proprioceptive, somatosensory, and vestibular sensation [35], and is thought to have affected the improvement of gait ability. It is thought that the stride length was increased in the visual cue deprivation balance training group due to the improvement of the flexor efferent contraction ability in the mid-stance phase as the weight transfer ability of the affected side and the stability of the hip joint on the affected side were improved [36]. In addition, it was reported that the increase in walking speed was closely related to the increase in stride length [37]. That is, in this study, it is thought that the improvement in stride time and cadence improved the increase in stride length.

Balance training in a situation where vision is blocked promotes the activity of other tracts besides the visual tract for balance control [38,39]. In other words, it improves the plasticity of various neural connections in the central nervous system and stimulates the tract that is not activated after the onset of a stroke. For this reason, it is thought that this forms reorganization of the cerebral cortex and has a positive effect on balance and walking ability.

The limitations of this study are as follows: first, it was difficult to generalize the results to all stroke patients due to the small number of subjects. Research is needed to investigate the effectiveness of visual cue deprivation balance training with head control by increasing the number of subjects in the future. Second, the feedback on head control relied on direct feedback from the physical therapist. In future studies, there is a need for a study in which patients can head control through a feedback sensor device for head adjustment.

## 5. Conclusions

In conclusion, the results of this study suggest that visual cue deprivation balance training is effective for improving the balance and walking ability of stroke patients. In particular, visual cue deprivation balance training applying head control is effective in improving dynamic balance ability. Therefore, it is important to maintain head orientation when performing visual cue deprivation balance training. In other words, correct vestibular input is very important for maintaining balance. Therefore, it is necessary to constantly maintain the head orientation by feedback and to properly control the head movement.

## Figures and Tables

**Figure 1 medicina-58-00629-f001:**
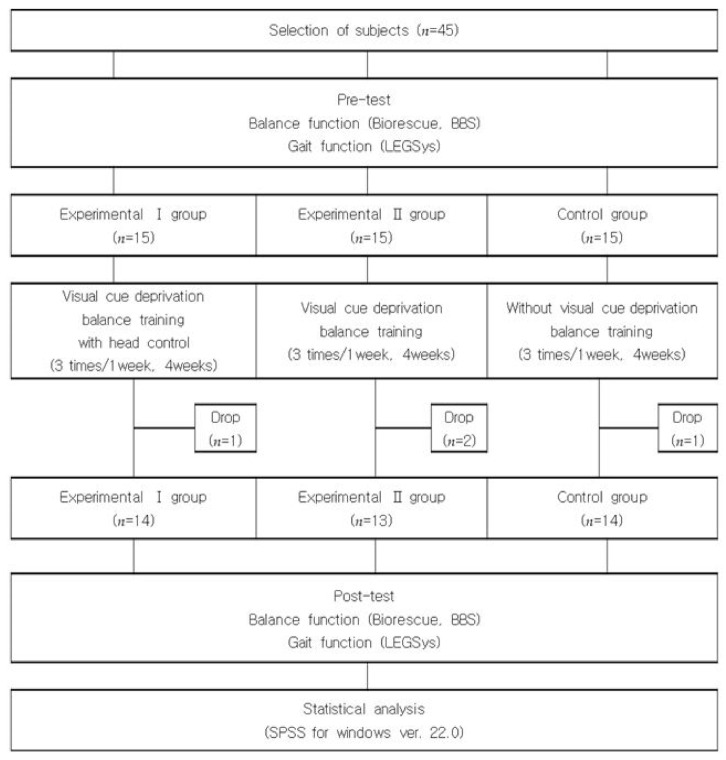
Diagram of experimental procedure.

**Figure 2 medicina-58-00629-f002:**
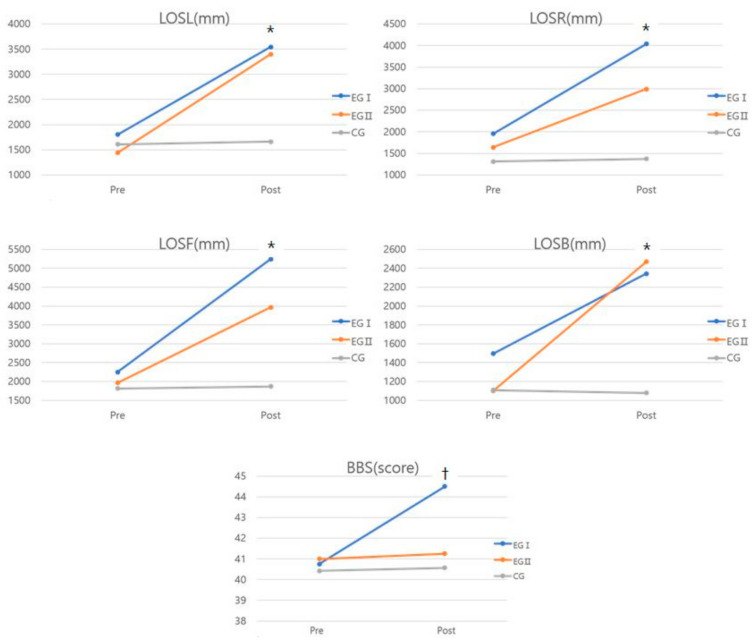
Comparison of balance function for each group. * Significant difference between the EGI and CG, EGII and CG. ^†^ Significant difference between the EGI and EGII, CG.

**Figure 3 medicina-58-00629-f003:**
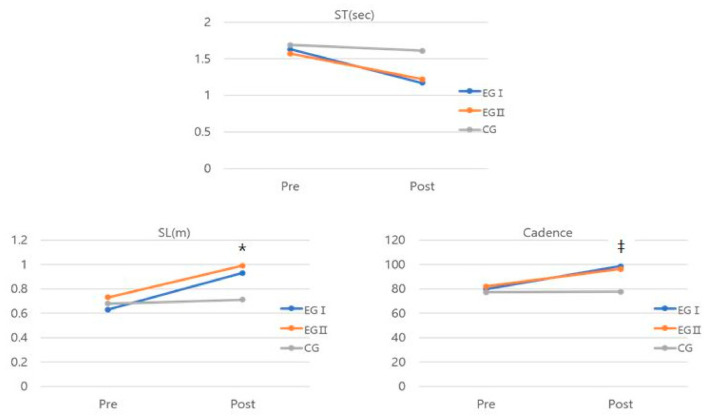
Comparison of gait function for each group. * Significant difference between the EGI and CG, EGII and CG. ^‡^ Significant difference between the EGI and CG.

**Table 1 medicina-58-00629-t001:** General characteristics of subjects.

	EGI (*n* = 14)	EGII (*n* = 13)	CG (*n* = 14)	*p*
Gender (Male/Female)	7/7	5/8	8/6	0.619
Age (year)	63.26 ± 12.4	68.01 ± 10.22	64.39 ± 11.55	0.708
Height (cm)	161.65 ± 13.72	154.09 ± 5.09	163.93 ± 9.27	0.178
Weight (kg)	63.32 ± 7.33	58.65 ± 9.21	63.65 ± 12.71	0.532
On set (month)	25.41 ± 25.32	23.30 ± 19.56	21.87 ± 19.32	0.689
Affected side (Left/Right)	6/8	7/6	9/5	0.522
Etiology (Hemorrhage/Infarction)	6/8	5/8	3/11	0.448

Mean ± SD: mean ± standard deviation. EGI; visual cue deprivation balance training with head control, EGII; visual cue deprivation balance training, CG; without visual cue deprivation balance training.

**Table 2 medicina-58-00629-t002:** Comparison of balance function for each group.

		EGI (*n* = 14) ^a^	EGII (*n* = 13) ^b^	CG (*n* = 14) ^c^	Group × Time		Post-hoc
					*F*	*η^2^*	
LOSL (mm)	Pre	1801.09 ± 969.98 (1240.75–2361.05)	1439.31 ± 633.83 (879.20–1999.42)	1608.69 ± 589.59 (1009.71–2207.67)	11.103 *	0.525	a, b > c
	Post	3544.31 ± 1801.69 (2487.10–4601.52)	3401.02 ± 1500.08 (2343.98–4458.06)	1662.82 ± 660.78 (532.71–2792.93)			
	*p*	0.002 *	0.001 *	0.578			
LOSR (mm)	Pre	1955.29 ± 824.37 (1463.59–2446.99)	1640.68 ± 707.59 (1149.02–2132.34)	1310.03 ± 322.891 (784.35–1835.71)	8.315 *	0.455	a, b > c
	Post	4040.52 ± 2042.62 (2954.04–5127.01)	2989.59 ± 1386.23 (1903.17–4076.01)	1372.25 ± 358.52 (210.54–2533.96)			
	*p*	0.003 *	0.005 *	0.055			
LOSF (mm)	Pre	2248.39 ± 976.38 (1647.50–2849.25)	1962.57 ± 858.31 (1361.88–2563.62)	1814.48 ± 490.77 (1172.07–2456.79)	9.057 *	0.475	a, b > c
	Post	5244.41 ± 2769.67 (3809.13–6679.62)	3967.52 ± 1709.42 (2532.25–5402.75)	1867.87 ± 517.09 (333.51–3402.20)			
	*p*	0.004 *	0.001 *	0.293			
LOSB (mm)	Pre	1495.78 ± 830.27 (1045.51–1946.26)	1099.53 ± 441.62 (649.11–1549.91)	1108.31 ± 459.07 (626.81–1589.79)	8.709 *	0.467	a, b > c
	Post	2342.29 ± 1165.36 (1580.38–3104.15)	2469.38 ± 1236.35 (1707.54–3231.25)	1078.16 ± 435.631 (263.67–1892.63)			
	*p*	0.004 *	0.005 *	0.475			
BBS (score)	Pre	40.76 ± 2.81 (38.10–43.41)	41.08 ± 2.28 (38.61–43.89)	40.42 ± 4.65 (37.74–43.41)	20.433 *	0.672	a > b, c
	Post	44.53 ± 2.55 (42.09–46.91)	41.27 ± 2.75 (38.61–43.39)	40.58 ± 4.96 (37.88–42.98)			
	*p*	0.000 *	0.749	0.604			

Mean ± SD: mean ± standard deviation. EGI; visual cue deprivation balance training with head control, EGII; visual cue deprivation balance training, CG; without visual cue deprivation balance training, LOSL; Limit of Stability Left, LOSR; Limit of Stability Right, LOSF; Limit of Stability Forward, LOSB; Limit of Stability Back; a, b, c means differences between groups, * *p* > 0.05.

**Table 3 medicina-58-00629-t003:** Comparison of gait function for each group.

		EGI (*n* = 14) ^a^	EGII (*n* = 13) ^b^	CG (*n* = 14) ^c^	Group × Time		Post-hoc
					** *F* **	** *η* ** ** ^2^ **	
ST (sec)	Pre	1.64 ± 0.59 (1.27–2.01)	1.57 ± 0.79 (1.22–1.94)	1.69 ± 0.48 (1.31–2.08)	2.967	0.230	
	Post	1.16 ± 0.39 (0.95–1.41)	1.22 ± 0.37 (1.00–1.44)	1.61 ± 0.52 (1.39–1.85)			
	*p*	0.016 *	0.014 *	0.139			
SL (m)	Pre	0.63 ± 0.33 (0.48–0.78)	0.73 ± 0.87 (0.59–0.89)	0.68 ± 0.21 (0.55–0.87)	17.608 *	0.638	a, b > c
	Post	0.93 ± 0.53 (0.80–1.07)	0.99 ± 0.38 (0.86–1.13)	0.71 ± 0.83 (0.55–0.83)			
	*p*	0.000 *	0.003 *	0.341			
Cadence	Pre	79.77 ± 18.65 (64.67–94.87)	81.98 ± 23.01 (66.90–97.08)	77.28 ± 20.65 (61.15–93.42)	3.953 *	0.286	a > c
	Post	98.71 ± 11.45 (87.36–110.06)	96.27 ± 5.01 (84.92–107.63)	77.57 ± 8.17 (65.43–89.71)			
	*p*	0.011 *	0.041 *	0.705			

Mean ± SD: mean ± standard deviation. EGI; visual cue deprivation balance training with head control, EGII; visual cue deprivation balance training, CG; without visual cue deprivation balance training, BBS; Berg Balance Scale, ST; Stride Time, SL; Stride Length, a, b, c means differences between groups, * *p* > 0.05.

## Data Availability

The datasets used during the current study are available from the corresponding author on reasonable request.

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
