# Peer review of "Effects of Visual Cue Deprivation Balance Training with Head Control on Balance and Gait Function in Stroke Patients"

_medicina, 2022, doi:10.3390/medicina58050629_

Round 1
Reviewer 1 Report
The article in the introduction and discussion is re-theoretized. The introduction is too long and there is little interesting (non-academic) background information from the research.
In part - the results, the authors should not refer to the table (see line 168, 175, 180), but should concentrate on discussing the results.
In part, the discussion is very little polemic with other authors, despite the fact that this section is quite long.
References . Out of 38 items, only 1/3 are less than 15 years old.
Author Response
Dear Reviewer, we sincerely appreciate your insightful comments. We have answered each of your points below.
- The article in the introduction and discussion is re-theoretized. The introduction is too long and there is little interesting (non-academic) background information from the research.
Author response: Thank you for your feedback. We removed the parts that were not needed in the introduction, and partially modified the overlaps with the discussion. Please, check again line 26-27, 39-41, 64-67, 246-249.
- In part - the results, the authors should not refer to the table (see line 168, 175, 180), but should concentrate on discussing the results.
Author response: Thank you for your detail advice. We revisedd results part.
- In part, the discussion is very little polemic with other authors, despite the fact that this section is quite long.
Author response: Thank you for your comments. We have partially revised the discussion part. Please, check again line 246-249, 259-260, 273-278.
- References . Out of 38 items, only 1/3 are less than 15 years old.
Author response: We have updated and revised as much as possible with the latest reference (ref. no. 1, 2, 3, 5, 8).
Reviewer 2 Report
The study appears to be well conducted and the topic is interesting. The statistical analysis carried out is detailed and solid. Within the study there are some grammatical errors such as line 265 change “res1ult” in “result”. At line 294 change “Therefor” in “therefore”. Please give a look to the English style.
What about the allocation concealment?
Why did you not plan to also evaluate the sense of position of the lower limb? It may be an interesting element that would have supported the results more strongly.
Author Response
Dear Reviewer, we sincerely appreciate your insightful comments. We have answered each of your points below.
- The study appears to be well conducted and the topic is interesting. The statistical analysis carried out is detailed and solid. Within the study there are some grammatical errors such as line 265 change “res1ult” in “result”. At line 294 change “Therefor” in “therefore”. Please give a look to the English style.
Author response: Thank you for your favorable feedback. We revised it. Please, check again this part.
- What about the allocation concealment?
Author response: Thank you for your comments. We wrote the contents of this part in line 103-106. If the part that needs to be revised in my paper does not fit your opinion direction, I will try to fix it as much as possible, so please let me know the details.
- Why did you not plan to also evaluate the sense of position of the lower limb? It may be an interesting element that would have supported the results more strongly.
Author response: Thank you for your detail advice. It was difficult for us to measure the part about the sense of position. The measurement using a manual goniometer has a severe error. For this reason, a digital goniometer was needed, but it could not be measured for environmental reason. We sincerely appreciate your comprehension.